# Effect of Annealing Temperature on the Structural and Electrochemical Properties of Hydrothermally Synthesized NiCo_2_O_4_ Electrodes

**DOI:** 10.3390/nano14010079

**Published:** 2023-12-27

**Authors:** Seok-Hee Lee, Hyun-Jin Cha, Junghwan Park, Chang-Sik Son, Young-Guk Son, Donghyun Hwang

**Affiliations:** 1School of Materials Science and Engineering, Pusan National University, Busan 46241, Republic of Korea; leesh91@pusan.ac.kr (S.-H.L.); jk260df@pusan.ac.kr (H.-J.C.); dhrms3306@pusan.ac.kr (J.P.); 2Division of Materials Science and Engineering, Silla University, Busan 46958, Republic of Korea; csson@silla.ac.kr

**Keywords:** hydrothermal synthesis, supercapacitor, pseudo-capacitor, energy storage system, annealing temperature

## Abstract

In this study, a porous Ni-foam support was employed to enhance the capacitance of nickel cobaltite (NiCo_2_O_4_) electrodes designed for supercapacitors. The hydrothermal synthesis method was employed to grow NiCo_2_O_4_ as an active material on Ni-foam. The NiCo_2_O_4_ sample derived from hydrothermal synthesis underwent subsequent post-heat treatment at temperatures of 250 °C, 300 °C, and 350 °C. Thermogravimetric analysis of the NiCo_2_O_4_ showed that weight loss due to water evaporation occurs after 100 °C and enters the stabilization phase at temperatures above 400 °C. The XRD pattern indicated that NiCo_2_O_4_ grew into a spinel structure, and the TEM results demonstrated that the diffraction spots (DSs) on the (111) plane of the sample annealed at 350 °C were more pronounced than those of other samples. The specific capacitance of the NiCo_2_O_4_ electrodes exhibited a decrease with increasing current density across all samples, irrespective of the annealing temperature. The electrode annealed at 350 °C recorded the highest specific capacitance value. However, the capacity retention rate of the NiCo_2_O_4_ electrode revealed a deteriorating trend, declining to 88% at 250 °C, 75% at 300 °C, and 63% at 350 °C, as the annealing temperature increased.

## 1. Introduction

Electrochemical energy storage and conversion devices are considered to be one of the most important power sources of renewable and alternative energy because they are designed to be sustainable and environmentally friendly in energy generation [1,2]. Among these devices, batteries and supercapacitors are classified as efficient systems that can act as power sources. High-energy-density batteries and high-power-density supercapacitors are being applied as power sources for portable electronic devices, electric vehicles, unmanned aerial vehicles, etc. [3,4]. In particular, supercapacitors are emerging as energy storage devices that can improve the shortcomings identified with batteries due to their fast charge/discharge characteristics, high power density, and excellent stability [5].

Supercapacitors are classified into three types based on their storage mechanism: electric double-layer (EDLC) capacitors, hybrid capacitors, and pseudo-capacitors. EDLC capacitors consist of two porous carbon electrodes separated by a liquid electrolyte of salts dissolved in aqueous or organic solvents, and use a mechanism to store energy through the adsorption and desorption of ions [6]. However, EDLC capacitors have limitations in realizing the high specific capacitance characteristics required by the industry. The hybrid capacitor, an electrochemical energy storage device, amalgamates the Faradaic rechargeable pseudo-capacitor electrode with the non-Faradaic reversible capacitor electrode. This integration of technologies harnesses the benefits inherent to both capacitor types, resulting in an enhancement of electrochemical properties [7]. Pseudo-capacitors represent a mechanism for storing electrical energy through redox reactions on the surface of electrode materials. In addition to transition metals, a diverse range of materials, including polymers, oxides, nitrides, sulfides, and phosphides, is utilized as electrodes for pseudo-capacitors [8,9,10,11,12,13]. Within the spectrum of energy storage materials, MnO_2_ presents appealing attributes, including high theoretical capacitance, eco-friendly characteristics, and cost-effectiveness. However, realizing its full theoretical capacitance is impeded by its intrinsic poor electrical conductivity [14]. MoO_3_, as an abundant resource, holds promise for robust electrochemical energy storage in batteries or supercapacitors, but its low electronic conductivity and limited ion diffusion capabilities present challenges in achieving theoretical capacitance [15]. Despite exhibiting high specific capacitance values, RuO_2_ encounters commercial limitations due to its elevated cost and inherent toxicity concerns [16].

Pseudo-capacitors are anticipated to exhibit significant capacitance and can be manufactured on a cost-effective large scale due to environmentally friendly production processes and low material costs, rendering them suitable as sustainable energy storage solutions. However, inherent drawbacks, such as compromised conductivity and frequent aggregation, arise due to the incomplete structure. To address this structural limitation, stable crystal growth of the electrode material can be facilitated through hydrothermal synthesis. As materials for pseudo-capacitors, Co oxide and Ni oxide are mechanically excellent and highly stable, prompting numerous studies. However, since the desired capacitance level cannot be achieved with individual oxides alone, an electrode material that utilizes the synergistic effect of the two oxides by appropriately mixing Co oxide and Ni oxide is applied. Cobalt oxide (Co_3_O_4_) and nickel cobaltite (NiCo_2_O_4_) are representative mixtures of these two oxides and have similar crystal structures. NiCo_2_O_4_ has been reported to exhibit superior electrical properties compared to Co_3_O_4_ [17]. Given the characteristics of capacitors, energy storage occurs through reactions on the surface; thus, a design that maximizes the surface area is crucial. With these design considerations, NiCo_2_O_4_ electrodes are crafted into nanostructured shapes, including sheets, wires, rods, flowers, and microspheres [18]. Among these shapes, the nanoflower boasts the largest surface area but suffers from a retention disadvantage. In comparison, among sheet and wire types with structures that are tailored based on the concentration of OH group-generating additives without the addition of various other compounds, nanosheets exhibit superior mechanical and electrical properties. An effective approach to enhance the capacitance of capacitors is by utilizing a conductive support as a substrate for the growth of the active material. Representative conductive supports, such as nickel foam (Ni-foam), nickel foil (Ni-foil), and nickel mesh (Ni-mesh), are commonly employed to improve the electrical properties of capacitors. Among them, Ni-foam has the largest specific surface area and its electrical properties are also reported to be about three times better than Ni-mesh [5,19]. 

While NiCo_2_O_4_ exhibits promising characteristics as a supercapacitor electrode material, achieving optimal performance requires careful consideration of synthesis and processing conditions. The utilization of a porous Ni-foam support offers a substantial surface area for active material growth, thereby potentially enhancing capacitance. However, the synthesis process may introduce resistance issues due to the binders used, which can adversely affect the overall performance of the supercapacitor. To address this challenge and promote active material growth on Ni-Foam, we employed a hydrothermal synthesis method with the addition of hexamethylenetetramine (HMT). HMT, upon decomposition, releases a significant amount of OH^−^ ions. These OH^−^ ions react with Ni^2+^ and Co^2+^ in the nucleus, facilitating the formation of mixed NiCo-hydroxide [20]. This process is beneficial for the growth of NiCo_2_O_4_ material, potentially improving the electrical and electrochemical properties of the supercapacitor. Following synthesis, the NiCo_2_O_4_ samples were subjected to post-heat treatment at various temperatures (250 °C, 300 °C, and 350 °C) to further optimize their properties. The selection of the annealing temperature significantly impacts the thermal properties, crystallographic characteristics, specific surface area, and stoichiometric compositions of the resulting NiCo_2_O_4_ electrodes. Therefore, we systematically investigated these factors using a range of characterization techniques, including TGA-DSC, XRD, TEM, BET, and XPS. Furthermore, the micro-surface morphology of the NiCo_2_O_4_ electrode, which can also influence the electrode’s performance, was observed using FE-SEM. Finally, the electrical properties of the electrodes, determining the supercapacitor’s performance, were identified through CV, GCD, and EIS analyses.

## 2. Experimental Section

### 2.1. Synthesis of NiCo_2_O_4_ Active Materials and Electrode Preparation 

The nickel foam (Ni-foam), utilized as the support, underwent a sequence of surface treatment processes. Initially, the Ni-foam was sonicated in a 1.0 M hydrochloric acid (HCl, Samchun, Seoul, Republic of Korea) solution for 1 h to remove contaminants and nickel oxide (NiO) attached to the surface. Subsequently, it was continuously sonicated in acetone, absolute ethanol, and distilled water for 30 min each, resulting in the achievement of a highly clean surface. The treated nickel foam was subjected to vacuum drying at 60 °C for a duration of 12 h. To prepare a solution for hydrothermal synthesis, a mixture comprising 3.5 mL of ethanol (Daejung, Siheung-si, Republic of Korea) diluted in 31.5 mL of distilled water was initially created. Subsequently, 2 mmol of cobalt (Ⅱ) nitrate hexahydrate (Co(NO_3_)_2_·6H_2_O, Junsei, Tokyo, Japan), 4 mmol of nickel (Ⅱ) nitrate hexahydrate (Ni(NO_3_)_2_·6H_2_O, Junsei, Tokyo, Japan), and 7 mmol of hexamethylenetetramine (HMT, Sigma Aldrich, Seoul, Republic of Korea) were added. The resulting solution underwent magnetic stirring for 1 h at room temperature. The previously washed nickel foam and the stirred mixed solution were transferred to a 50 mL Teflon bottle and then assembled in a hydrothermal synthesizer. The assembled synthesizer was then placed in an oven and heated at 120 °C for 6 h to facilitate the synthesis of NiCo_2_O_4_ active materials on the nickel foam. The synthesized samples underwent a gradual cooling process to room temperature, followed by washing with distilled water and ethanol and, finally, drying in an oven at 70 °C for 12 h. The dried samples were placed in an oven and post-heat treated in an air atmosphere for 2 h at temperatures of 250 °C, 300 °C, and 350 °C. 

### 2.2. Characterizations of NiCo_2_O_4_ Active Materials and Electrodes

The thermal properties of the NiCo_2_O_4_ active materials were analyzed over a temperature range from 25 °C to 800 °C using a thermogravimetric analyzer–differential scanning calorimeter (TGA-DSC, STA 8000, PerkinElmer, Waltham, MA, USA). The growth orientation and crystal structure of each sample were measured by X-ray diffraction (XRD, SmartLab, Rigaku, Tokyo, Japan) using Cu K_α_ radiation (λ = 1.541 Å) within a 2θ range of 5° to 80°. The microstructure and crystallography of NiCo_2_O_4_ were characterized using transmission electron microscopy (TEM, JEM-2100F, JEOL, Tokyo, Japan). The specific surface area, pore size, and volume of the sample were determined using the Brunauer-Emmett-Teller (BET, ASAP 2010, Micromeritics, Norcross, GA, USA) method. The elemental composition of the samples and the electronic states of the atoms in the material were examined using X-ray photoelectron spectroscopy (XPS, K-Alpha, Thermo Fisher Scientific, Seoul, Republic of Korea). The surface morphology of the NiCo_2_O_4_ electrodes was observed using field-emission scanning electron microscopy (FE-SEM, MIRA3, Tescan Korea, Seoul, Republic of Korea). 

To analyze the electrochemical properties of the NiCo_2_O_4_ electrodes, a three-electrode system was established, comprising a reference electrode (Hg/HgO), a counter electrode (Pt), and a working electrode (NiCo_2_O_4_). The electrolyte used was 1 M potassium hydroxide (KOH, Junsei, Tokyo, Japan) solution. The mass loading of the NiCo_2_O_4_ electrodes, fabricated at annealing temperatures of 250 °C, 300 °C, and 350 °C, were determined to be 1.580 mg/cm^2^, 1.039 mg/cm^2^, and 0.731 mg/cm^2^, respectively. Utilizing data collected from an electrochemical workstation (ZIVE MP2A, WonATech, Seoul, Republic of Korea) connected to the system, cyclic voltammetry (CV), galvanostatic charge–discharge (GCD), and electrochemical impedance spectroscopy (EIS) were applied to investigate the effect of annealing temperatures on the properties. 

## 3. Results and Discussion

### 3.1. Thermal Properties of NiCo_2_O_4_ Active Materials 

Figure 1 presents the results of the thermal properties of a NiCo_2_O_4_ active material using a thermogravimetric analyzer–differential scanning calorimeter (TGA-DSC). Changes in the weight loss and endothermic–exothermic reactions of the NiCo_2_O_4_ samples were measured in a temperature range from room temperature (RT) to 800 °C under atmospheric conditions. The dashed line in the TGA graph of Figure 1a indicates the moisture and weight loss within NiCo_2_O_4_ before reaching a specific temperature. The weight loss rates for the samples heat-treated at temperatures of 250 °C, 300 °C, and 350 °C were 1.1%, 24.4%, and 53.1%, respectively. The weight loss curve of the sample passes 350 °C and enters the stabilization stage in the section above 400 °C, indicating the completion of sample crystallization at 350 °C, where the weight loss rate is the highest. The mechanism associated with crystallization due to weight loss is shown in chemical equation 1 [21]. The DSC graph in Figure 1b reveals the exothermic reaction, indicating that after crystallization of NiCo_2_O_4_ is complete at 350 °C, decomposition into NiO and Co_2_O_3_ proceeds. The reactions occurring at 300 °C and 350 °C, which are the crystallization temperatures (T_c_), are expressed by chemical equations 2 [22].
(1)$$2\mathrm{Co}(\mathrm{OH}{)}_{2}+\mathrm{Ni}(\mathrm{OH}{)}_{2}+\frac{1}{2}\;O_{2}\to {\mathrm{NiCo}}_{2}O_{4}+{3H}_{2}O$$
NiCo_2_O_4_ → NiO + Co_2_O_3_(2)

### 3.2. Structural Properties of NiCo_2_O_4_ Active Materials 

Figure 2 presents the results of the X-ray diffraction (XRD) analysis, depicting the crystal structure of the hydrothermally synthesized NiCo_2_O_4_ active materials. The diffraction peaks of the NiCo_2_O_4_ sample were observed at 2θ angles of 18.7°, 31.4°, 36.7°, 37.7°, 44.4°, 55.6°, 58°, 64.9°, and 77.3°. The peaks indicated the (111), (220), (311), (222), (400), (422), (511), (440), and (533) planes, respectively. The XRD pattern analysis revealed a spinel crystal structure (JCPDS No. 20-0781), with a preferential growth orientation along the (311) plane for the hydrothermally synthesized NiCo_2_O_4_ [23]. No discernible changes in crystal structure or phase were observed with varying annealing temperatures [24]. 

The full width at half maximum (FWHM) of the primary diffraction peak decreased to 1.152 at 250 °C, 0.768 at 300 °C, and 0.288 at 350 °C. The crystallite size of the NiCo_2_O_4_ samples was calculated by substituting the FWHM value obtained from XRD data into Scherrer’s formula [25]. The calculated crystallite size at each annealing temperature increased by 7.6 nm at 250 °C, 11.4 nm at 300 °C, and 30.6 nm at 350 °C, indicating an enhancement in crystallinity with higher annealing temperatures. 

Figure 3 illustrates the TEM analysis results of NiCo_2_O_4_ samples annealed from 250 °C to 350 °C. The top panel shows low-magnification TEM images and the scale bars are 100 nm. The middle panel displays the electron diffraction pattern (EDP) images, and the bottom panel presents magnified EDP images. The specimens fabricated by hydrothermal synthesis were polycrystals and, through EDP analysis, a ring pattern corresponding to the crystal growth orientation of each specimen was obtained. In the middle panel, the EDP illustrates the crystal orientation of NiCo_2_O_4_ with a white dashed line, while the crystal orientation of NiO is denoted by a yellow dashed line. The specimen annealed at 250 °C exhibited solely blurred diffraction spots corresponding to the (111) plane of NiO. In the 300 °C specimen, blurred spots corresponding to both the (111) and (200) planes were observed, but the shapes of the spots remained indistinct and could not be easily distinguished. In the 350 °C specimen, spots indicating the (111) and (200) planes were also present, but the diffraction spots of the (111) plane appeared more distinct compared to those at other temperatures. This indicates that crystalline NiO precipitated from the NiCo_2_O_4_ matrix during the annealing process at 350 °C. The TEM analysis results were consistent with the thermal properties previously described and chemical equation 2 of NiCo_2_O_4_.

### 3.3. BET Analysis of NiCo_2_O_4_ Active Materials

Surface area and pore volume are acknowledged to be critical factors influencing the characteristics of capacitor materials. Brunauer–Emmett–Teller (BET) gas adsorption measurements were utilized to determine both BET and Barrett–Joyner–Halenda (BJH) factors [26].

As shown in Table 1, the pore size of NiCo_2_O_4_ at each annealing temperature was determined to be 3.6 nm at 250 °C, 4.0 nm at 300 °C, and 5.3 nm at 350 °C, indicating a mesoporous nature [27]. The surface area of the sample at each temperature was measured to be 258.4 m^2^/g at 250 °C, 129.7 m^2^/g at 300 °C, and 148.8 m^2^/g at 350 °C. Concurrently, the pore volume increased from 0.51 cm^3^/g to 0.77 cm^3^/g as the annealing temperature escalated from 250 °C to 350 °C. These findings affirm that, with the elevation in annealing temperature, the surface area decreases, while the pore volume and size comparatively increase. The results of the NiCo_2_O_4_ surface area analysis conducted through BET/BJH measurements are depicted in Figure 4. The BET/BJH hysteresis loop observed for each sample exhibits a type IV loop, indicative of the filling and emptying of mesopores through capillary condensation [28].

### 3.4. Chemical Properties of NiCo_2_O_4_ Active Materials

Figure 5 presents the chemical analysis of NiCo_2_O_4_ active materials annealed at 350 °C. In Figure 5a, the comprehensive X-ray photoelectron spectroscopy (XPS) spectrum of NiCo_2_O_4_ reveals well-defined peaks corresponding to Ni 2p, Co 2p, O 1s, and C 1s. The presence of C 1s signifies carbon contamination introduced during experimental procedures or equipment usage. In Figure 5b, the Ni 2p spectrum of the sample at 350 °C is elaborated, revealing prominent peaks at 854.5 eV and 871.95 eV, corresponding to Ni 2p_3/2_ and Ni 2p_1/2_ states, with a separation of 17.45 eV. Each peak is accompanied by two shake-up satellite peaks at 861.1 eV and 879.5 eV. The peaks at 853.6 eV and 871.1 eV denote the Ni^2+^ state, while the peaks at 855.4 eV and 872.8 eV represent the Ni^3+^ state [29]. Two clearly identifiable peaks, Ni^2+^ and Ni^3+^, are observed on the Ni 2p spectrum. The spin-orbit splitting energy difference (ΔE = 2p_1/2_ − 3p_3/2_) was calculated to be 17.5 eV for Ni^2+^ 2p_3/2_ and 17.4 eV for Ni^3+^ 2p_3/2_ [30]. 

The Co 2p spectrum shown in Figure 5c exhibits two primary peaks at 779.85 eV and 794.9 eV, separated by 15.05 eV in the Co 2p_3/2_ and Co 2p_1/2_ states. Each main peak is succeeded by three shake-up satellite peaks at 782.6, 788.8, and 804.2 eV. The peaks at 779.2 eV and 794.3 eV correspond to the Co^3+^ state, while the peaks at 780.5 eV and 795.5 eV represent the Co^2+^ state. Figure 5d shows the O 1s spectrum featuring three distinct oxygen peaks. The primary peak of M-O-M is observed at 529.1 eV, signifying a metal-oxygen bond, while the -OH peak emerges at 530.7 eV, indicative of a multiple defect site with low oxygen coordination. The H-OH peak represents the physical and chemical bonds of residual moisture present on the material surface [31].

### 3.5. Morphological Properties of NiCo_2_O_4_ Electrodes

Figure 6 shows FE-SEM images of the surface morphology of NiCo_2_O_4_ electrodes grown through hydrothermal synthesis on nickel foam (Ni-foam). Figure 6a–c presents low-magnification views of the samples annealed at different temperatures. Well-established growth of NiCo_2_O_4_ active materials throughout the Ni-foam was confirmed. Figure 6d–f provides high-magnification images, unveiling intricate details of the surface morphology of NiCo_2_O_4_ active materials synthesized on the Ni-foam surface. At an annealing temperature of 250 °C, the electrodes grow in the form of nanopillars and nanowires. The surface shape undergoes deformation with an increase in annealing temperature. The structures at 300 °C and 350 °C exhibit nanosheet microstructures with a wavy, silk-like appearance [32]. Nanosheet arrays and porous nanostructures grown on Ni-foam promote charge transport and ion diffusion without binder blocks [33].

### 3.6. Electrochemical Properties of NiCo_2_O_4_ Electrodes

A comprehensive investigation and analysis were conducted to explore the capacitance properties of three electrode samples prepared through hydrothermal synthesis, followed by annealing at different temperatures. Cyclic voltammetry (CV) data, collected over a potential range of 0 V to 0.6 V at various scan rates ranging from 10 mV to 50 mV, are summarized in Figure 7. The samples distinctly exhibit redox characteristics typical of pseudo-capacitors. An oxidation reaction occurs between 0.4 V and 0.55 V, followed by a reduction reaction between 0.2 V and 0.4 V. The observed magnitude of the current response indicates robust Faradaic response characteristics of battery-type electrode materials [34]. The Faraday reaction, which represents the interaction of electrical properties, can be expressed by the following chemical equation [35,36,37]: NiCo_2_O_4_ + OH^−^ → NiOOH + 2CoOOH + e^−^(3)
CoOOH + OH^−^ → CoO_2_ + H_2_O +e^−^(4)
NiO + OH^−^ → NiOOH + e^−^(5)

A pair of redox peaks is observed at the NiCo_2_O_4_ electrodes, originating from a Faradaic reaction involving M-O/M-O-OH (where M represents Ni and Co ions) associated with the anion OH^−^ [38]. The two pairs of redox current peaks correspond to the reversible reactions of Co^2+^/Co^3+^ and Ni^2+^/Ni^3+^ transitions [39]. The appearance of additional peaks, distinct from Ni^2+^/Ni^3+^ and Co^2+^/Co^3+^, in the CV plot at lower scan rates may be attributed to the presence of an oxygen-containing compound, NiO [40]. Two oxidation peaks, labeled A_1_ and A_2_, and reduction peaks, denoted C_1_ and C_2_, were identified [41,42]. With higher annealing temperatures, the dominance of the A_2_ peak over A_1_ and the prevalence of the C_2_ peak compared to C_1_ become more pronounced. This trend correlates with the NiO precipitation observed in the TEM images.

Electrochemical measurements were conducted using the galvanostatic charge–discharge (GCD) method on three electrodes under various current densities, while maintaining a limiting potential of 550 mV. The three graphs depicted in Figure 8 show consistent reversible patterns. During the charge–discharge process, a non-linear graph expressing the unique charge–discharge characteristics of a battery-type pseudo-capacitor was observed, rather than the typical linear curve seen with an electric double-layer capacitor (EDLC) [43]. The sample annealed at 250 °C exhibits an ideal graph for charging and discharging. The sample annealed at 350 °C displays a flat graph, requiring approximately 450 s for charging. However, during discharging, all the stored energy is released in about 150 s due to a rapid voltage drop. The specific capacitance is calculated using the following equation [44]: C = (I × ∆t)/(m × ∆V)(6)

In this equation, C (F/g) represents the specific capacitance, I (A) is the current during the discharge process, ∆t (s) is the discharge time, ∆V (V) is the electromotive force, and m (g) is the mass of the active material. The specific capacitance of the NiCo_2_O_4_ electrode annealed at 250 °C was determined to be 371 F/g, 351 F/g, 330 F/g, 311 F/g, 287 F/g, and 248 F/g under varying current densities. Notably, there was a discernible decrease in specific capacitance with the escalation of current density. The reduction in capacity with rising current density and scanning speed can be attributed to the contact time with the active surface of the material and the extent of ionic contact [45]. The specific capacitance of the electrode annealed at 300 °C, identified by the exothermic reaction and crystallization temperature in DSC analysis (in Figure 1), was calculated to be 893 F/g, 798 F/g, 750 F/g, 710 F/g, 662 F/g, and 605 F/g at respective current densities. The specific capacitance of the NiCo_2_O_4_ electrode annealed at 350 °C, where the exothermic reaction and crystallization reached a stable state, was determined to be 1254 F/g, 1075 F/g, 1038 F/g, 987 F/g, 898 F/g, and 866 F/g for each corresponding current density. This GCD analysis confirms that the annealing temperature of 350 °C serves as a significant factor in enhancing the charge–discharge time of the NiCo_2_O_4_ electrode. The energy density and power density values were calculated using the equation E = 1/2·(C·V^2^), where E represents energy density, C is the specific capacitance, and V is the voltage [46]. The NiCo_2_O_4_ electrode annealed at 250 °C displayed an energy density of 56 W/g, while the electrode annealed at 300 °C exhibited a density of 135 W/g, and the electrode annealed at 350 °C demonstrated a density of 188 W/g. The power density of the electrodes was determined with the formula P = E/Δt, where P denotes power density, E is energy density, and Δt is the discharge time [47]. All three NiCo_2_O_4_ electrodes (250 °C, 300 °C, 350 °C) exhibited a power density of 0.275 W/g.

Figure 9a illustrates Nyquist plots obtained from electrochemical impedance spectroscopy (EIS), which measures impedance. The real impedance (Z’) is mathematically expressed as Z’ = R_s_ + R_ct_, where R_s_ denotes electrolyte resistance and R_ct_ signifies charge transfer resistance of electrode [48]. EIS measurements offer valuable insights into the resistance characteristics of each electrode. The NiCo_2_O_4_ electrode annealed at 250 °C demonstrates an R_s_ value of 0.86 Ω. The electrodes annealed at 300 °C and 350 °C exhibit R_s_ values of 0.57 Ω and 0.60 Ω, respectively. The R_ct_ values were determined through the fitting process of the equivalent circuit using the equation R_s_ + Cd1/(R_ct_ + W) + C_ps_. The presented equivalent circuit consists of electrode internal resistance (R_s_), Faraday charge transfer resistance (R_ct_), diffusion resistance (W), Faradaic pseudocapacitance (C_ps_), and a constant phase element (Cdl) [49]. The R_ct_ of the electrode annealed at 300 °C decreased to 1.018 Ω, while the R_ct_ value for the electrode annealed at the higher temperature of 350 °C showed a slight increase to 1.348 Ω. The Nyquist plot deviates from the typical semicircular shape, demonstrating a relatively linear configuration. This suggests that the oxidation-substrate interface resistance is minimized, nearly eliminating the Nyquist plot [50]. These results mean that the interfacial resistance between the Ni-foam and NiCo_2_O_4_ active material is minimized, and suggest that the contact resistance problem can be solved by adding HMT without using a binder.

In Figure 9b, the variation in capacitance per current density is presented. All three electrodes consistently demonstrated a gradual decrease in capacitance with increasing current density. The electrode sample annealed at 350 °C exhibited a higher rate of capacitance decrease compared to the one annealed at 250 °C. In Figure 9c, the capacitance retention rates for each sample were measured at a scan rate of 7 A/g. The electrode sample annealed at 250 °C exhibited an 88% retention rate, while the 300 °C sample showed a retention rate of 75%. The sample annealed at 350 °C demonstrated a retention rate of 63%. 

## 4. Conclusions

In this study, we aimed to enhance the capacitance of NiCo_2_O_4_ electrodes for supercapacitors using a porous Ni-foam support. The synthesis process involved a hydrothermal method for growing NiCo_2_O_4_ on Ni-foam, with the addition of HMT to address resistance issues arising from binders during synthesis. NiCo_2_O_4_ samples underwent post-heat treatment at varying temperatures of 250 °C, 300 °C, and 350 °C. The impact of annealing temperatures on the structural and electrochemical properties of NiCo_2_O_4_ samples was analyzed. The thermal properties revealed that weight loss due to water evaporation occurred post 100 °C, stabilizing beyond 450 °C. The weight loss rates for the samples subjected to heat treatment at temperatures of 250 °C, 300 °C, and 350 °C were 1.1%, 24.4%, and 53.1%, respectively. The XRD pattern confirmed the NiCo_2_O_4_ active material, which exhibited a spinel structure, and the TEM results highlighted sharper diffraction spots on the NiO (111) plane for the sample annealed at 350 °C. The FE-SEM image shows the formation of nanopillar and nanowire shapes at an annealing temperature of 250 °C. Subsequently, this surface morphology transformed into a nanosheet microstructure with a silk-like appearance at 300 °C and 350 °C. The analysis of CV results revealed that the NiCo_2_O_4_ electrodes, prepared through hydrothermal synthesis, exhibited typical redox characteristics indicative of a pseudo-capacitor. Moreover, it demonstrated Faraday response properties suitable for application as a battery-type electrode material. Despite a general decrease in the specific capacitance with increasing current density across all samples, the NiCo_2_O_4_ electrode annealed at 350 °C exhibited the highest specific capacitance. However, the capacity retention rate demonstrated a decline, reaching 88% at 250 °C, 75% at 300 °C, and 63% at 350 °C.

## Figures and Tables

**Figure 1 nanomaterials-14-00079-f001:**
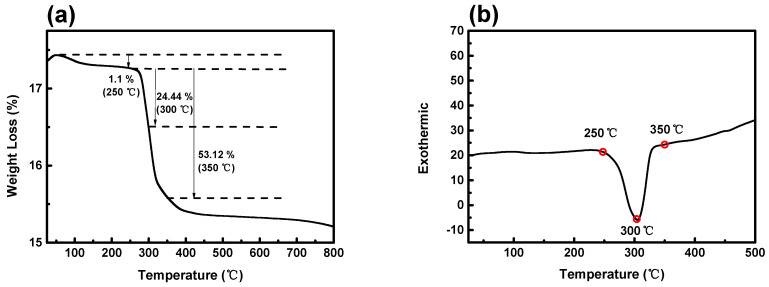
(**a**,**b**) TGA-DSC profile of the NiCo_2_O_4_ sample as a function of temperatures.

**Figure 2 nanomaterials-14-00079-f002:**
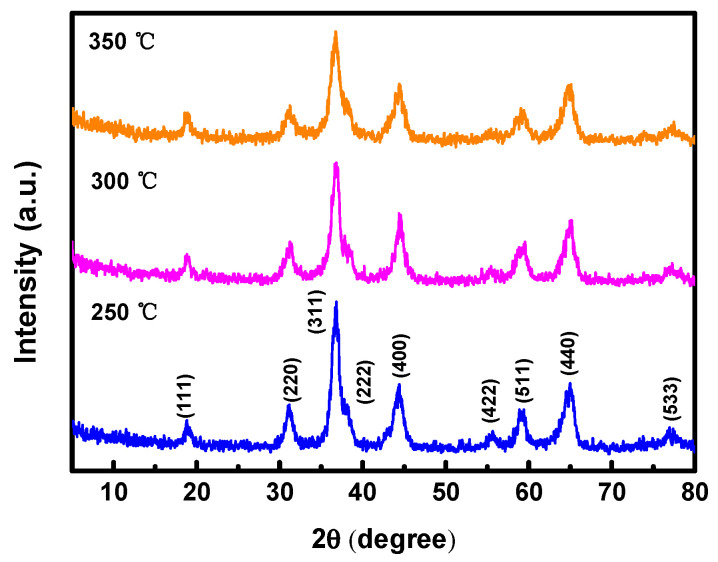
XRD spectra of NiCo_2_O_4_ samples prepared by annealing at different temperatures.

**Figure 3 nanomaterials-14-00079-f003:**
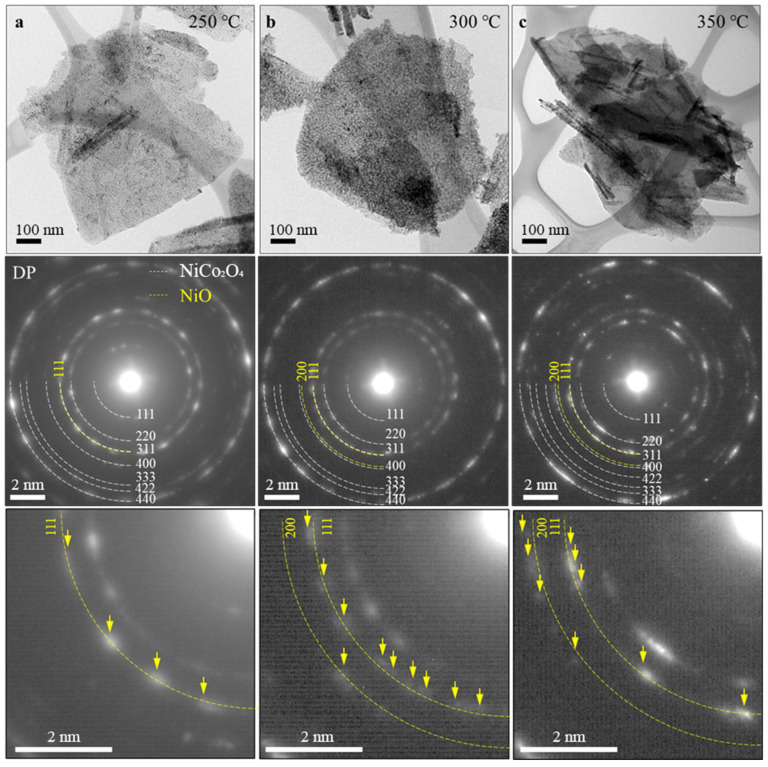
TEM analysis results of NiCo_2_O_4_ samples annealed from 250 °C to 350 °C: (top panel) low-magnification TEM images, (middle panel) EDP images, and (bottom panel) magnified EDP images.

**Figure 4 nanomaterials-14-00079-f004:**
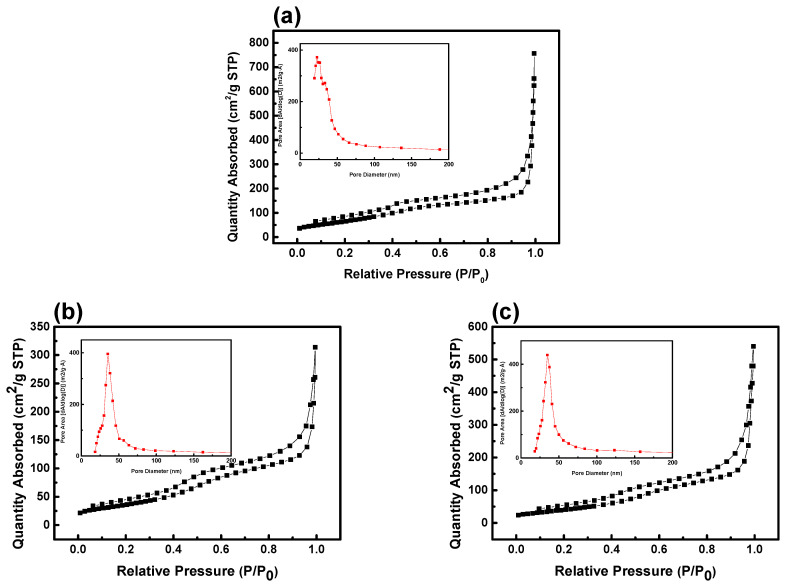
BET and BJH results for NiCo_2_O_4_ samples at different annealing temperatures: (**a**) 250 °C, (**b**) 300 °C, and (**c**) 350 °C.

**Figure 5 nanomaterials-14-00079-f005:**
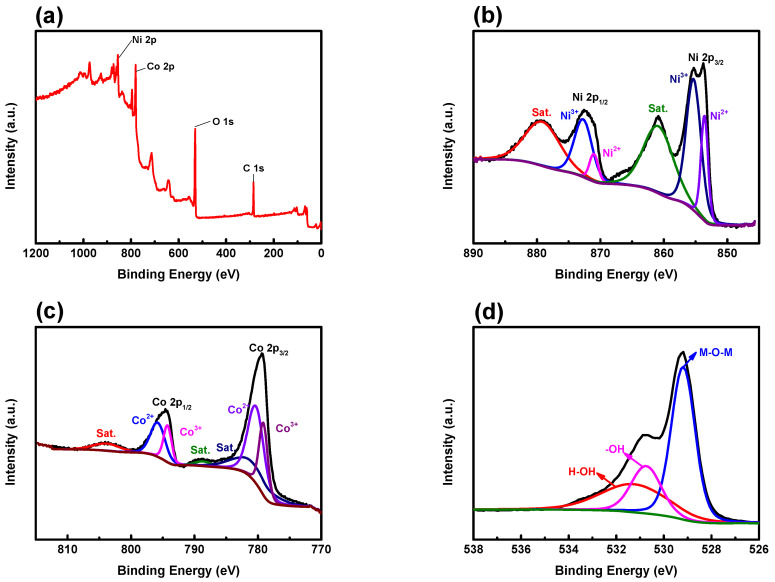
XPS Spectrum of NiCo_2_O_4_ samples annealed at 350 °C: (**a**) full spectrum, (**b**) Ni 2p, (**c**) Co 2p, and (**d**) O 1s.

**Figure 6 nanomaterials-14-00079-f006:**
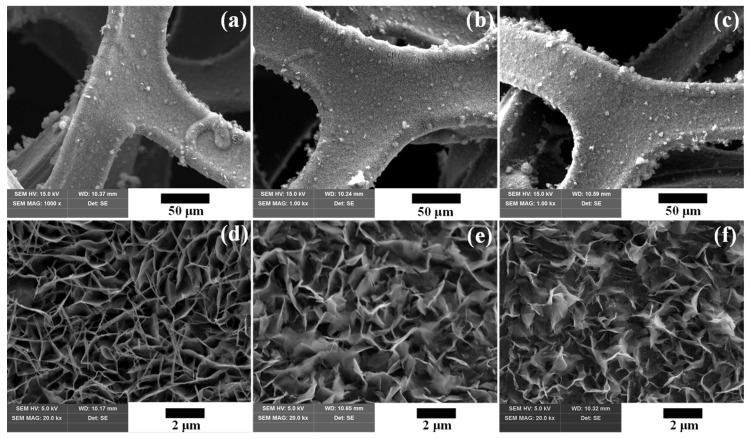
High- and low-magnification FE-SEM images of NiCo_2_O_4_ electrode surfaces prepared by annealing at various temperatures: (**a**,**d**) 250 °C, (**b**,**e**) 300 °C, and (**c**,**f**) 350 °C.

**Figure 7 nanomaterials-14-00079-f007:**
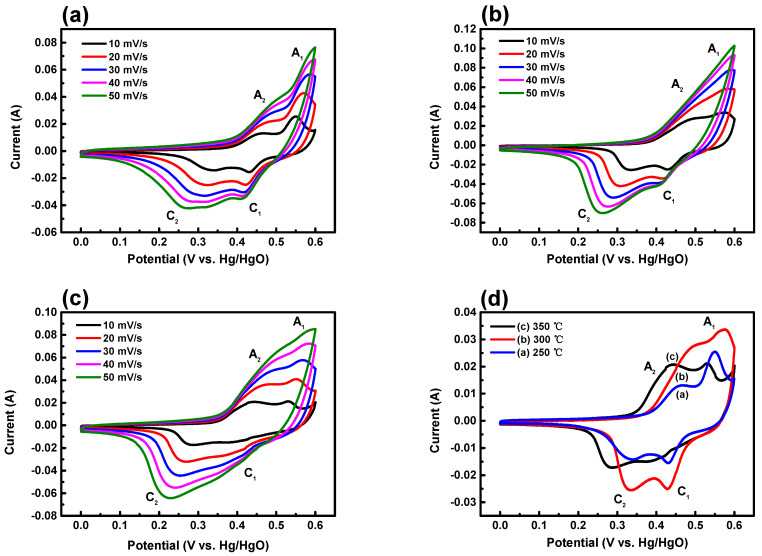
CV profiles of NiCo_2_O_4_ electrodes annealed at different temperatures: (**a**) 250 °C, (**b**) 300 °C, (**c**) 350 °C, and (**d**) comparison of CV values at 10 mV.

**Figure 8 nanomaterials-14-00079-f008:**
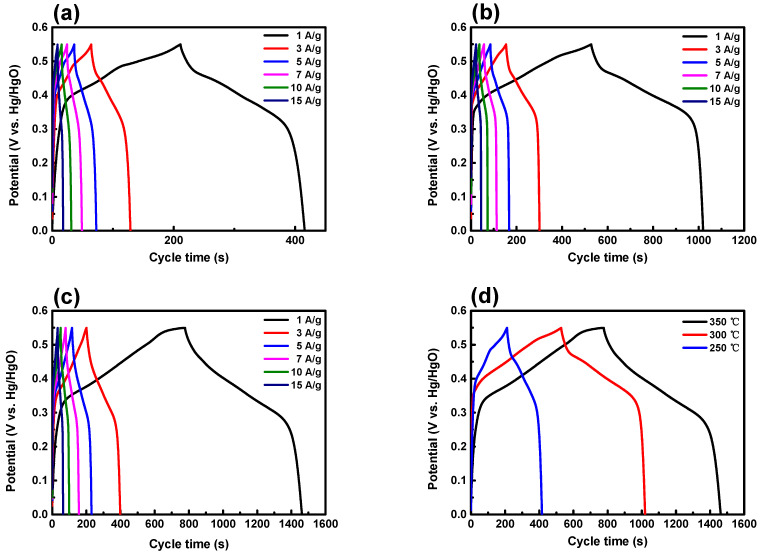
GCD profiles of NiCo_2_O_4_ electrodes annealed at different temperatures: (**a**) 250 °C, (**b**) 300 °C, (**c**) 350 °C, and (**d**) comparison of GCD values in 1 A/g.

**Figure 9 nanomaterials-14-00079-f009:**
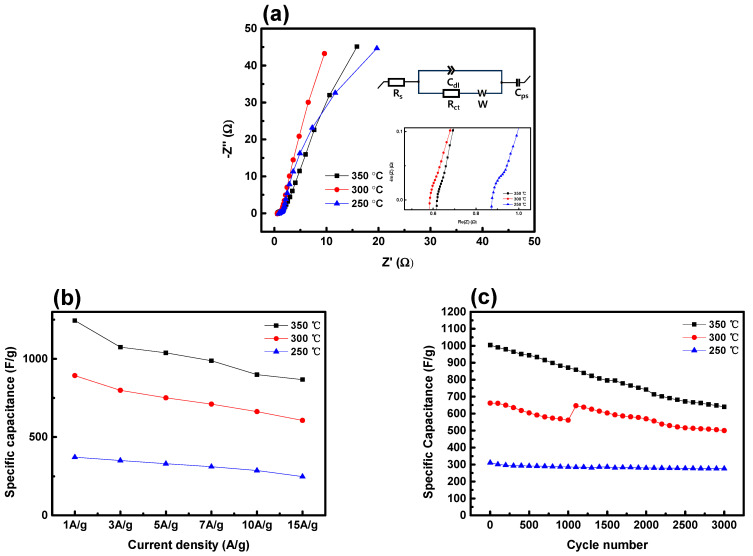
(**a**) EIS analysis of NiCo_2_O_4_ electrodes, (**b**) specific capacitance at various current densities, and (**c**) capacitive retention as a function of cycle number.

**Table 1 nanomaterials-14-00079-t001:** BET/BJH results of NiCo_2_O_4_ active materials according to various annealing temperatures.

Temperature (°C)	Surface Area (m^2^/g)	Pore Volume (cm^3^/g)	Pore Size (nm)
250	258.44	0.505	3.645
300	129.69	0.413	3.995
350	148.76	0.773	5.258

## Data Availability

Data are contained within the article.

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
