# Peer review of "Effect of Annealing Temperature on the Structural and Electrochemical Properties of Hydrothermally Synthesized NiCo2O4 Electrodes"

_nanomaterials, 2023, doi:10.3390/nano14010079_

Round 1

Reviewer 1 Report

Comments and Suggestions for Authors

Research article by Seok-Hee Lee et al. as “Influence of Annealing Temperature on the Structural and Electrical Characteristics of Hydrothermally Synthesized NiCo2O4 Electrodes” employed the hydrothermal synthesis method to grow NiCo2O4 on nickel foam. The prepared electrode exhibits satisfactory electrochemical performance. After carefully reading, it was found that this article needs major revision because several issues and explanations are still need to be clarified. I recommend it publication in this journal after providing proper improvement in revised version by including suggestion, modification and reply to raised queries which are given below.

1.     Revise the abstract to succinctly elucidate the work and the obtained results. Additionally, incorporate the tested data in the application section.

2.     Nowadays, various energy storage devices have been developed based on versatile nanomaterials. It would be better to give a short comparison on the advantages and disadvantages of those devices. Some typical references are suggested to be cited to enrich the content, such as Journal of Alloys and Compounds 2022, 903, 163824; Rare Metals 2022, 41 (10), 3432-3445.

3.     Please provide clarification on the source of raw materials used for material preparation.

4.     The electrode preparation process and the loading mass of individual electrodes should be supplemented.

5.     The scale above Figure 3 should be annotated with specific numerical values corresponding to the lengths indicated.

6.     Attention should be given to the issue of superscripts and subscripts in numerical values, particularly in Figure 5.

7.     The format issue regarding the abbreviation of "Figure" in the text should be standardized.

8.     Attention should be paid to the spacing between numbers and units in the manuscript.

9.     The current in the CV curves in the manuscript is not normalized. Is the loading mass of the electrodes used in each test the same?

10.  The scan rate unit for the CV curves should be "mV/s" instead of "mV".

11.  The numerical intervals on the horizontal and vertical axes of the EIS analysis plot in Figure 9a should be equal in length. Please refer and cite Journal of Energy Storage 2024, 77, 109859.

12.  The formatting of brackets in the manuscript should be standardized.

13.  The textual discussion in the captions of Figures 8 and 9 does not correctly correspond to the respective images.

14.  Please ensure uniformity in the format of references, particularly with regard to the journal names.

Comments on the Quality of English Language

Moderate editing of English language is required.

Author Response

Dear Reviewer and Editor,

We appreciate your kind consideration and the insightful comments on our recent submission (Manuscript ID: nanomaterials-2782418). Based on your opinion, we have corrected the manuscript and resubmit the revised version. We hope this revision can meet the reviewer’s requirements and accept the paper for publication. The followings are our point-by-point responses:

 #. Reviewer’s comments & Response

Comment 1. Revise the abstract to succinctly elucidate the work and the obtained results. Additionally, incorporate the tested data in the application section.

Response: The abstract was revised to explain the research results more concisely. The tested data was integrated into the sections presented by the reviewer.

 Comment 2. Nowadays, various energy storage devices have been developed based on versatile nanomaterials. It would be better to give a short comparison on the advantages and disadvantages of those devices. Some typical references are suggested to be cited to enrich the content, such as Journal of Alloys and Compounds 2022, 903, 163824; Rare Metals 2022, 41 (10), 3432-3445.

Response: Content describing the advantages and disadvantages of energy storage devices based on various nanomaterials has been added to the introduction section. This modified content can be found on lines 52 to 59.

Comment 3. Please provide clarification on the source of raw materials used for material preparation.

Response: The exact source of raw materials used for material preparation has been added in lines 111 to 121 in the experimental section.

Comment 4. The electrode preparation process and the loading mass of individual electrodes should be supplemented.

Response: The electrode preparation process and the loading mass of individual electrodes were modified to reflect the opinions provided by the reviewer. The modified content can be found in the experimental section, lines 145 to 150.

 Comment 5. The scale above Figure 3 should be annotated with specific numerical values corresponding to the lengths indicated.

Response: The scale above Figure 3 has been revised to include specific numerical values corresponding to the indicated lengths.

 Comment 6. Attention should be given to the issue of superscripts and subscripts in numerical values, particularly in Figure 5.

Response: The superscripts and subscripts of the numerical values shown in Figure 5 were re-examined and any parts displayed incorrectly were corrected.

 Comment 7. The format issue regarding the abbreviation of "Figure" in the text should be standardized.

Response: We observed variations in the use of "Figures" and "Fig" in the text and have standardized them to "Figure" for consistency. The necessary corrections have been made throughout the document.

Comment 8. Attention should be paid to the spacing between numbers and units in the manuscript.

Response: In this manuscript, all parts where the spacing between numbers and units was incorrectly written have been corrected.

Comment 9. The current in the CV curves in the manuscript is not normalized. Is the loading mass of the electrodes used in each test the same?

Response: “The appearance of additional peaks, distinct from Ni2+/Ni3+ and Co2+/Co3+, in the CV plot at lower scan rates may be attributed to the presence of an oxygen-containing compound, NiO [40]. Two oxidation peaks, labeled A1 and A2, and reduction peaks, denoted C1 and C2, were identified [41, 42]. With higher annealing temperatures, the dominance of the A2 peak over A1 and the prevalence of the C2 peak compared to C1 become more pronounced. This trend correlates with the NiO precipitation observed in the TEM images.” This modified content can be found on lines 296 to 302.

The mass loading of the electrode used in CV testing depends on the annealing conditions. The mass loading of the NiCo2O4 electrodes, fabricated at annealing temperatures of 250 °C, 300 °C, and 350 °C, were determined to be 1.580 mg/cm2, 1.039 mg/cm2, and 0.731 mg/cm2, respectively. The mass loading of the different electrodes was specified in the experimental section.

Comment 10. The scan rate unit for the CV curves should be "mV/s" instead of "mV".

Response: The scan rate unit of the CV profile in Figure 7 was modified to “mV/s”.

Comment 11. The numerical intervals on the horizontal and vertical axes of the EIS analysis plot in Figure 9a should be equal in length. Please refer and cite Journal of Energy Storage 2024, 77, 109859.

Response: The numerical intervals on the horizontal and vertical axes of Figure 9(a) were corrected by referring to the reference paper presented

Comment 12. The formatting of brackets in the manuscript should be standardized.

Response: The parenthetical format of the manuscript has been standardized.

Comment 13. The textual discussion in the captions of Figures 8 and 9 does not correctly correspond to the respective images.

Response: The explanations for Figures 8 and 9 have been modified to exactly match the corresponding images.

Comment 14. Please ensure uniformity in the format of references, particularly with regard to the journal names.

Response: The notation format of references, including the journal name, was modified to match the format presented in “Nanomaterials”.

All the comments pointed out by the reviewer have been revised and the revised text is marked in red for easy identification.

Very truly yours,

Donghyun Hwang

Reviewer 2 Report

Comments and Suggestions for Authors

Although the authors provided many characterization of three samples, it is hard to clarify the reason why 300oC NiCo2O4 shows the highest capacitance. It would be better to offer some conclusion or discussion to explain this point.

In addition, some qustions are as listed below:

1. In the abstract, “To mitigate resistance issues attributed to the binder during the synthesis process, hexamethylenetetramine (HMT) was introduced” is stated. This sentence is not completely correct, since to mitigate resistance, the authors developed flower-like NiCo2O4. It’s better to make the abstract clear.

2. The units in figure 7 are not correct. It should be mV/s

3. Why is the potential window different in CV curves and GCD curves?

4. In line329, it should be figure 9 if the author refer to Nyquist plot.

5. Why is the 300 oC NiCo2O4 has the lowest resistance but fail to provide high capacitance or the highest retention?

6. In figure 9c, why there is an inconsistency in 300oC NiCo2O4?

Comments on the Quality of English Language

The quility of English is okay. However, the abstract should be improved.

Author Response

Dear Reviewer and Editor,

We appreciate your kind consideration and the insightful comments on our recent submission (Manuscript ID: nanomaterials-2782418). Based on your opinion, we have corrected the manuscript and resubmit the revised version. We hope this revision can meet the reviewer’s requirements and accept the paper for publication. The followings are our point-by-point responses:

 #. Reviewer’s comments & Response

Comment 1. In the abstract, “To mitigate resistance issues attributed to the binder during the synthesis process, hexamethylenetetramine (HMT) was introduced” is stated. This sentence is not completely correct, since to mitigate resistance, the authors developed flower-like NiCo2O4. It’s better to make the abstract clear.

Response: To make the abstract more clear, the previously described sentence “To mitigate resistance issues attributed to the binder during the synthesis process, hexamethylenetetramine (HMT) was introduced” was deleted.

 Comment 2. The units in figure 7 are not correct. It should be mV/s

Response: The scan rate unit of the CV profile in Figure 7 was modified to “mV/s”.

Comment 3. Why is the potential window different in CV curves and GCD curves?

Response: The difference in settings between the CV potential window and the GCD potential window is attributed to the following reasons: The CV potential window is defined based on the range of the maximum oxidation peak observed, and the range is expressed in the GCD as the electrode's capacity. The exclusion of the potential window beyond the maximum oxidation peak in GCD was done to optimize electrode efficiency.

Comment 4. In line329, it should be figure 9 if the author refer to Nyquist plot.

Response: The Nyquist plot schematized based on EIS measurement data was modified to Figure 9(a). In addition, the parts incorrectly marked as Figure 8(b) and 8(c) in the subsequent explanation were corrected to Figure 9(b) and Figure 9(c).

 Comment 5. Why is the 300 oC NiCo2O4 has the lowest resistance but fail to provide high capacitance or the highest retention?

Response: TEM analysis in Figure 3 revealed the precipitation of NiO at annealing temperatures of 300 °C and 350 °C. The presence of NiO is believed to enhance conductivity and specific capacitance, as reported in (doi:http://doi.org/10.1016/j.jcis.2017.05.027). The Rs value for the electrode annealed at 250 °C was 0.86 ohm. Conversely, electrodes prepared at 300 °C and 350 °C exhibited lower Rs values of 0.57 ohm and 0.60 ohm, respectively. The specific capacitance was also confirmed to have higher values at 300 °C and 350 °C, which are the temperatures at which NiO precipitates, compared to 250 °C. However, the retention rate was most stable at 88% at 250 °C, decreasing to 75% at 300 °C and 63% at 350 °C with increasing annealing temperature. Based on these results, it was found that NiO can have a positive effect on improving conductivity and specific capacitance and a negative effect on retention rate. Despite these findings, the specific cause of the lower specific capacitance at 300 °C compared to 350 °C remains inconclusive, lacking quantitative measurements or specific evidence.

We intend to delve deeper into understanding this phenomenon in future research. Your insights and advice have been incredibly valuable, and We are grateful for your generosity in sharing your time and knowledge with us.

 Comment 6. In figure 9c, why there is an inconsistency in 300 oC NiCo2O4?

Response: The discrepancy observed in Figure 9(c) is attributed to experimental error. The graph is constructed from data obtained by measuring the electrode six times for 500 cycles each. It is presumed that an error related to the y-axis range occurred during the data collection process.

All the comments pointed out by the reviewer have been revised and the revised text is marked in red for easy identification.

Very truly yours,

Donghyun Hwang

Reviewer 3 Report

Comments and Suggestions for Authors

The materials such as NiO and Co3O4 have been widely investigated due to their low cost and good electrochemical activity. However, due to the lack of high conductivity, mixed metal oxides along with carbon composites are widely considered as electrodes for energy storage applications including supercapacitors. In the submitted work by Seok-Hee Lee et al, the effect of annealing temperature on the characteristic properties and electrochemical performance of nickel cobaltite synthesised on porous Ni-foam support was investigated and compared their capacitance values. It is summarised that storage properties decline at a higher annealing temperature. The reported materials and its approach on annealing temperature is well known but maybe new for the chosen supercapacitor application.

The work is well presented, and the manuscript is OK with reasonable physical and electrochemical data drawn from the SEM/ TEM morphological images and surface properties. The work in its present form is publishable but needs some revisions before rendering a final decision. 

The following points need to be considered.

·         The title is too long.

·         Was the effect of the Ni/Co ratio on the electrochemical properties of the system with the annealing temperature?

·         What is the role of HMT? It is unclear.

·         On page 1, line 40; actually, supercapacitors are classified into three including hybrid capacitors as reported by Minakshi et al, modify and discuss.

·         Please provide the values for energy density and power density for nickel cobaltite.

·         Section 3.5 must be called Electrochemical properties.

·         The shape of the CV curve and their capacitance values must be compared to similar materials like Co3O4 (doi.org/10.3390/nano7110356); and Co compounds (doi.org/10.1002/cplu.201600294). The reason for the sharp decline in capacitance retention observed in Fig. 9b must be explained.

·         Is there any data for the electrode system?

·         The Rct values must be depicted from the EIS plot.

·         The equivalent circuit must be provided in Figure 9a.

·         The section's conclusion is too long.

·         What is the mass of the electrode? Is this a freestanding electrode?

Comments on the Quality of English Language

Language is fine with some tweaks/polish is required.

Author Response

Dear Reviewer and Editor,

We appreciate your kind consideration and the insightful comments on our recent submission (Manuscript ID: nanomaterials-2782418). Based on your opinion, we have corrected the manuscript and resubmit the revised version. We hope this revision can meet the reviewer’s requirements and accept the paper for publication. The followings are our point-by-point responses:

 #. Reviewer’s comments & Response

Comment 1. The title is too long.

Response: The previous title “Influence of Annealing Temperature on the Structural and Electrical Characteristics of Hydrothermally Synthesized NiCo2O4 Electrodes” has been revised to “Effect of Annealing Temperature on the Structural and Electrochemical Properties of Hydrothermally Synthesized NiCo2O4 Electrodes”.

 Comment 2. Was the effect of the Ni/Co ratio on the electrochemical properties of the system with the annealing temperature?

Response: This study did not primarily focus on the effect of the Ni/Co ratio concerning annealing temperature, and consequently, variations in the Ni/Co ratio were neither controlled nor highlighted. The main objective was to explore the impact of annealing temperature on the synthesis and performance of NiCo2O4 electrodes.

Comment 3. What is the role of HMT? It is unclear.

Response: To provide a clear explanation of the HMT's role, the following has been added to the introduction section, found on lines 90 to 98: “However, the synthesis process may introduce resistance issues due to the binders used, which can adversely affect the overall performance of the supercapacitor. To address this challenge and promote active material growth on Ni-Foam, we employed a hydrothermal synthesis method with the addition of hexamethylenetetramine (HMT). The HMT, upon decomposition, releases a significant amount of OH- ions. These OH- ions react with Ni2+ and Co2+ in the nucleus, facilitating the formation of mixed NiCo-hydroxide [20]. This process is beneficial for the growth of NiCo2O4 material, potentially improving the electrical and electrochemical properties of the supercapacitor.”

Comment 4. On page 1, line 40; actually, supercapacitors are classified into three including hybrid capacitors as reported by Minakshi et al, modify and discuss.

Response: We revised the classification of supercapacitors into three categories, including hybrid capacitors, as reported by Minakshi et al. The modified content can be found on lines 39 to 48.

 Comment 5. Please provide the values for energy density and power density for nickel cobaltite.

Response: A paragraph describing the energy density and power density values of nickel cobaltite has been added from lines 334 to 341.

Comment 6. Section 3.5 must be called Electrochemical properties.

Response: The title of Section 3.5 has been revised to ‘Electrochemical Properties of NiCo2O4 Electrodes’.

 Comment 7. The shape of the CV curve and their capacitance values must be compared to similar materials like Co3O4 (doi.org/10.3390/nano7110356); and Co compounds (doi.org/10.1002/cplu.201600294). The reason for the sharp decline in capacitance retention observed in Fig. 9b must be explained.

Response: “The appearance of additional peaks, distinct from Ni2+/Ni3+ and Co2+/Co3+, in the CV plot at lower scan rates may be attributed to the presence of an oxygen-containing compound, NiO [40]. Two oxidation peaks, labeled A1 and A2, and reduction peaks, denoted C1 and C2, were identified [41, 42]. With higher annealing temperatures, the dominance of the A2 peak over A1 and the prevalence of the C2 peak compared to C1 become more pronounced. This trend correlates with the NiO precipitation observed in the TEM images.” This modified content can be found on lines 296 to 302.

TEM analysis in Figure 3 revealed the precipitation of NiO at annealing temperatures of 300 °C and 350 °C. The presence of NiO is believed to enhance conductivity and specific capacitance, as reported in (doi:http://doi.org/10.1016/j.jcis.2017.05.027). The Rs value for the electrode annealed at 250 °C was 0.86 ohm. Conversely, electrodes prepared at 300 °C and 350 °C exhibited lower Rs values of 0.57 ohm and 0.60 ohm, respectively. The specific capacitance was also confirmed to have higher values at 300 °C and 350 °C, which are the temperatures at which NiO precipitates, compared to 250 °C. However, the retention rate was most stable at 88% at 250 °C, decreasing to 75% at 300 °C and 63% at 350 °C with increasing annealing temperature. Based on these results, it was found that NiO can have a positive effect on improving conductivity and specific capacitance and a negative effect on retention rate. Despite these findings, the specific cause of the lower specific capacitance at 300 °C compared to 350 °C remains inconclusive, lacking quantitative measurements or specific evidence.

We intend to delve deeper into understanding this phenomenon in future research. Your insights and advice have been incredibly valuable, and We are grateful for your generosity in sharing your time and knowledge with us.

Comment 8. Is there any data for the electrode system?

Response: “To analyze the electrochemical properties of the NiCo2O4 electrodes, a three-electrode system was established, comprising a reference electrode (Hg/HgO), a counter electrode (Pt), and a working electrode (NiCo2O4). The electrolyte used was 1 M potassium hydroxide (KOH, Junsei, Tokyo, Japan) solution. The mass loading of the NiCo2O4 electrodes, fabricated at annealing temperatures of 250 °C, 300 °C, and 350 °C, were determined to be 1.580 mg/cm2, 1.039 mg/cm2, and 0.731 mg/cm2, respectively.” This modified content can be found on lines 145 to 150.

Comment 9. The Rct values must be depicted from the EIS plot.

Response: “The Rct values were determined through the fitting process of the equivalent circuit using the equation Rs + Cd1/(Rct + W) + Cps. The presented equivalent circuit consists of electrode internal resistance (Rs), Faraday charge transfer resistance (Rct), and diffusion resistance (W), Faradaic pseudocapacitance (Cps), and a constant phase element (Cdl) [49]. The Rct of the electrode annealed at 300 °C decreased to 1.018 Ω, while the Rct value for the electrode annealed at the higher temperature of 350 °C showed a slight increase to 1.348 Ω.” This modified content can be found on lines 353 to 359.

Comment 10. The equivalent circuit must be provided in Figure 9a.

Response: Figure 9(a) has been modified to display the equivalent circuit model (Rs + Cd1/(Rct + W) + Cps) used in the EIS analysis.

Comment 11. The section's conclusion is too long.

Response: The conclusion part has been edited for conciseness by deleting all unnecessary or lengthy content.

Comment 12. What is the mass of the electrode? Is this a freestanding electrode?

Response: The mass loading of the NiCo2O4 electrodes, fabricated at annealing temperatures of 250 °C, 300 °C, and 350 °C, were determined to be 1.580 mg/cm2, 1.039 mg/cm2, and 0.731 mg/cm2, respectively.

All the comments pointed out by the reviewer have been revised and the revised text is marked in red for easy identification.

Very truly yours,

Donghyun Hwang

Round 2

Reviewer 2 Report

Comments and Suggestions for Authors

The manuscript was well-revised.

Reviewer 3 Report

Comments and Suggestions for Authors

The revised version is suitable for publication. The authors have addressed my queries satisfactorily.